# A Novel Unified Architecture for Low-Shot Counting by Detection and Segmentation

**Jer Pelhan, Alan Lukežič, Vitjan Zavrtanik, Matej Kristan**
Faculty of Computer and Information Science, University of Ljubljana
jer.pelhan@fri.uni-lj.si

## Abstract

Low-shot object counters estimate the number of objects in an image using few or no annotated exemplars. Objects are localized by matching them to prototypes, which are constructed by unsupervised image-wide object appearance aggregation. Due to potentially diverse object appearances, the existing approaches often lead to overgeneralization and false positive detections. Furthermore, the best-performing methods train object localization by a surrogate loss, that predicts a unit Gaussian at each object center. This loss is sensitive to annotation error, hyperparameters and does not directly optimize the detection task, leading to suboptimal counts. We introduce GeCo, a novel low-shot counter that achieves accurate object detection, segmentation, and count estimation in a unified architecture. GeCo robustly generalizes the prototypes across objects appearances through a novel dense object query formulation. In addition, a novel counting loss is proposed, that directly optimizes the detection task and avoids the issues of the standard surrogate loss. GeCo surpasses the leading few-shot detection-based counters by $\sim25\%$ in the total count MAE, achieves superior detection accuracy and sets a new solid state-of-the-art result across all low-shot counting setups. The code is available on GitHub.

## 1 Introduction

Low-shot object counting considers estimating the number of objects of previously unobserved category in the image, given only a few annotated exemplars (few-shot) or without any supervision (zero-shot) [22]. The current state-of-the-art methods are predominantly based on density estimation [4; 14; 32; 26; 22; 31; 7; 31]. These methods predict a density map over the image and estimate the total count by summing the density.

While being remarkably robust for global count estimation, density outputs lack explainability such as object location and size, which is crucial for many practical applications [33; 30]. This recently gave rise to detection-based low-shot counters [20; 19; 35], which predict the object bounding boxes and estimate the total count as the number of detections. Nevertheless, detection-based counting falls behind the density-based methods in total count estimation, leaving a performance gap.

In detection-based counters, a dominant approach to identify locations of the objects in the image involves construction of object prototypes from few (e.g., three) annotated exemplar bounding boxes and correlating them with image features [20; 35; 19]. The exemplar construction process is trained to account for potentially large diversity of object appearances in the image, often leading to overgeneralization, which achieves a high recall, but is also prone to false positive detection. Post-hoc detection verification methods have been considered [20; 35] to address the issue, but their multi-stage formulation prevents exploiting the benefits of end-to-end training.

Currently, the best detection counters [20; 35] predict object locations based on the local maxima in the correlation map. During training, the map prediction is supervised by a unit Gaussian placed on

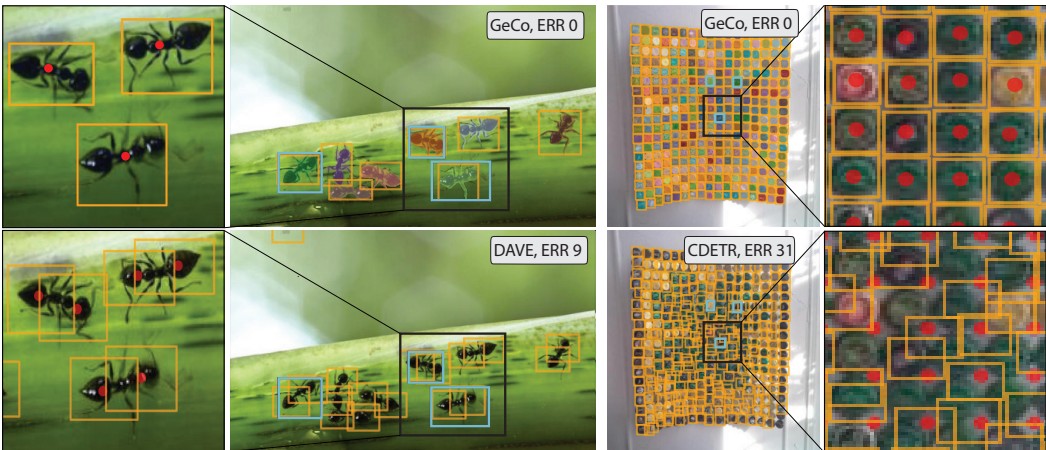

Figure 1: DAVE [20] predicts object centers (red dots) biased towards blob-like structures, leading to incorrect partial detections of ants (bottom left), while GeCo(ours) addresses this with the new loss (top left). CDETR [19] fails in densely populated regions (bottom right), while GeCo addresses this with the new dense query formulation by prototype generalization (top right). Exploiting the SAM backbone, GeCo delivers segmentations as well. Exemplars are denoted in blue.

each object center. However, the resulting surrogate loss is susceptible to the center annotation noise, requires nontrivial heuristic choice of the Gaussian kernel size and in practice leads to detection preference of compact blob-like structures (see Figure 1, column 1&2). Recently, DETR [1] inspired counter was proposed to avoid this issue [19], however, it fails in densely populated regions even though it applies a very large number of detection queries in a regular grid (see Figure 1, column 3&4).

We address the aforementioned challenges by proposing a new single-stage low-shot counter GeCo, which is implemented as an add-on network for SAM [12] backbone. A single architecture is thus trained for both few-shot and zero-shot setup, it enables counting by detection and provides segmentation masks for each of the detected objects. Our first contribution is a dense object query formulation, which applies a non-parametric model for image-wide prototype generalization (hence GeCo) in the encoder, and decodes the queries into highly dense predictions. The formulation simultaneously enables reliable detection in densely-populated regions (Figure 1, column 3&4) and prevents prototype over-generalization, leading to an improved detection precision at a high recall. Our second contribution is a new loss function for dense detection training that avoids the ad-hoc surrogate loss with unit Gaussians, it directly optimizes the detection task, and leads to improved detection not biased towards blob-like regions (Figure 1, column 1&2).

GeCo outperforms all detection-based counters on challenging benchmarks by 24% MAE and the density-based long-standing winner [4] by 27% MAE, while delivering superior detection accuracy. The method shows substantial robustness to the number of exemplars. In one-shot scenario, GeCo outperforms the best detection method in 5% AP50, 45% MAE and by 14% in a zero-shot scenario. GeCo is the first detection-based counter that outperforms density based counters in all measures by using the number of detections as the estimator, and thus sets a milestone in low-shot detection-based counting.

## 2 Related works

Traditional counting methods focus on predefined categories like vehicles[3], cells [5], people[15], and polyps, [33] requiring extensive annotated training data and lacking generalization to other categories, necessitating retraining or conceptual changes. Low-shot counting methods address this limitation by estimating counts for arbitrary categories with minimal or no annotations, enabling test-time adaptation.

With the proposal of the FSC147 dataset [24] low-shot counting methods emerged, which predict global counts by summing over a predicted density maps. The first method [24] proposed an

adaptation of a tracking backbone for density map regression. BMNet+ [26] tackled learning representation and similarity metric, while SAFECount [32] introduced a new feature enhancement module, improving appearance generalization. CounTR [14] utilized a vision transformer for image feature extraction and a convolutional network for encoding the exemplar features. LOCA [4] argued that exemplar shape information should be considered along with the appearance, and proposed an iterative object prototype extraction module. This led to a simplified counter architecture that remains a top-performer among density-based counters.

To improve explainability of the estimated counts and estimate object locations as well, detection-based methods emerged. The first few-shot detection-based counter [19] was an extended transformer-based object detector [2] with the ability to detect objects specified by the exemplars. Current state-of-the-art DAVE [20] proposed a two-stage detect-and-verify paradigm for low-shot counting and detection, wherein the first stage it generates object proposals with a high recall, but low precision, which is improved by a subsequent verification step. PSECO [35] proposed a three-stage approach called point-segment-and-count, which employs more involved proposal generation with better detection accuracy and also applies a verification step to improve precision. Both DAVE and PSECO are multi-stage methods that train a network for the surrogate task of predicting density maps for object centers, from which the bounding boxes are predicted. Although detection-based counters offer additional applicability, they fall behind the best density-based counters in global count estimation.

## 3   Single-stage low-shot object counting by detection and segmentation

Given an input image $I \in \mathbb{R}^{H_0 \times W_0 \times 3}$ and a set of $k$ exemplar bounding boxes $\boldsymbol{B}^{\mathrm{E}} = \{\mathbf{b}_i\}_{i=1:k}$ specifying the target category, the task is to predict bounding boxes $\boldsymbol{B}^P = \{\mathbf{b}_j\}_{j=1:N}$ for all target category objects in $I$, with the object count estimated as $N = |\boldsymbol{B}^P|$.

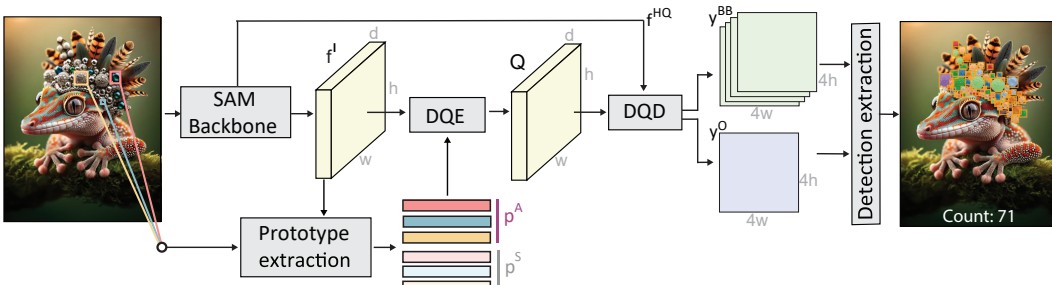

Figure 2: The architecture of the proposed single-stage low-shot counter GeCo.

The proposed detection-based counter GeCo pipeline proceeds as follows (see Figure 2). The image is encoded by a SAM [12] backbone into $\mathbf{f}^I \in \mathbb{R}^{h \times w \times d}$, where $h = H_0/r$, $w = W_0/r$ and $d$ is number of feature channels. In the few-shot setup, two kinds of prototypes (appearance and shape) are extracted from each annotated object exemplar. The appearance prototypes $\mathbf{p}^A \in \mathbb{R}^{k \times d}$ are extracted by RoI-pooling [9] features $\mathbf{f}^I$ from the exemplar bounding boxes. Following [4], shape prototypes $\mathbf{p}^S \in \mathbb{R}^{k \times d}$ are extracted as well, by $\mathbf{p}^S_i = \Phi([W_{\mathbf{b}_i}, H_{\mathbf{b}_i}])$, where $W_{\mathbf{b}_i}$ and $H_{\mathbf{b}_i}$ are the width and height of the $i$-th exemplar bounding box, and $\Phi(\cdot)$ is a small MLP network. The concatenation of $\mathbf{p}^A$ and $\mathbf{p}^S$ yields $\mathbf{p} \in \mathbb{R}^{2k \times d}$ prototypes.

Note, however, that in a zero-shot setup, exemplars are not provided and the task is to count the majority-class objects in the image. In this setup, a single zero-shot prototype is constructed by attending a pretrained objectness prototype $\mathbf{p}^Z$ to the image features, i.e., $\mathbf{p} = \mathrm{CA}(\mathbf{p}^Z, \mathbf{f}^I, \mathbf{f}^I)$, where $\mathrm{CA}(a, b, c)$ is cross-attention [28] followed by a skip connection, with $a$, $b$ and $c$ as attention query, key and value, respectively.

The prototypes $\mathbf{p}$ (either from few-shot or zero-shot setup) are then generalized across the image, and dense object detection queries are constructed by the Dense query encoder (DQE, Section 3.1). These are decoded into dense detections by the Dense query decoder (DQD, Section 3.2). The final detections are extracted and refined by a post-processing step (Section 3.3). The aforementioned modules are detailed in the following sections.

## 3.1 Dense object query encoder (DQE)

To account for the variation of the object appearances in the image, the current state-of-the-art [4; 20; 35] aims at constructing a small number of prototypes (e.g., three) that compactly encode the object appearance variation in the image, often leading to overgeneralization and false detections. We deviate from this paradigm by considering image-wide prototype generalization with a non-parametric model that constructs $w \cdot h$ location-specific prototypes $\mathbf{P}_{N_P} \in \mathbb{R}^{w \cdot h \times d}$. Let $\mathbf{P}_0 = \mathbf{f}^I$ be the initial dense generalized prototypes (i.e., one for each location). The final dense generalized prototypes $\mathbf{P}_{N_P}$ are calculated by the following iterative adaptation via cross-attention

$$\mathbf{P}_i = \mathrm{CA}(\mathbf{P}_{i-1}, \mathbf{p}, \mathbf{p}), \tag{1}$$

where $i \in \{1, ..., N_P\}$. Note that spatial encoding is not applied, to enable spatially-unbiased information flow from the prototypes $\mathbf{p}$ to all locations.

Next, dense object queries are constructed from the generalized prototypes by the following iterations

$$\mathbf{Q}_j = \mathrm{CA}(\mathrm{SA}(\mathbf{f}^I), \mathbf{Q}_{j-1}, \mathbf{Q}_{j-1}), \tag{2}$$

where $j \in \{1, ..., N_Q\}$, $\mathbf{Q}_0 = \mathbf{P}_{N_P}$, and $\mathrm{SA}(\cdot)$ is a self-attention followed by a skip connection to adapt the input features to the current queries. In both cross- and self-attentions, positional encoding is applied to enable location-dependent query construction. In the remainder of the paper, the dense object queries $\mathbf{Q}_{N_Q}$ are denoted as $\mathbf{Q}$ for clarity

## 3.2 Dense object query decoder (DQD)

The dense queries $\mathbf{Q}$ from Section 3.1 are decoded into object detections by a dense object query decoder (DQD). Note that the spatial reduction of image by the SAM backbone may lead to encoding several small objects into the same query in $\mathbf{Q}$. To address this, the object queries are first *spatially unpacked* into high-resolution dense object queries i.e., $\mathbf{Q}^{HR} \in \mathbb{R}^{H \times W \times d}$, where $H = H_0/2$, $W = W_0/2$ and $d$ is the number of feature channels. The unpacking process consists of three convolutional upsampling stages, with each stage composed of a $3 \times 3$ convolution, a Leaky ReLU and a $2\times$ bilinear upsampling. To facilitate unpacking of small objects, the features after the second stage are concatenated by the SAM-HQ features [11] $\mathbf{f}^{HQ}$ before feeding into the final stage.

Finally, the objectness score $\mathbf{y}^O \in \mathbb{R}^{H \times W \times 1}$ is calculated by a simple transform, i.e., $\mathbf{y}^O = \mathrm{LRelu}(\mathbf{W}_O \cdot \mathbf{Q}^{HR})$, where $\mathbf{W}_O$ is a learned projection matrix and $\mathrm{LReLU}(\cdot)$ is a Leaky ReLU. Each query is also decoded into the object pose by a three-layer MLP, i.e., $\mathbf{y}^{BB} = \sigma(\mathrm{MLP}(\mathbf{Q}^{HR}))$, where $\sigma(\cdot)$ is a sigmoid function and $\mathbf{y}^{BB} \in \mathbb{R}^{H \times W \times 4}$ are bounding box parameters in the *tlrb* format [27].

## 3.3 Detections extraction and refinement

The final detections are extracted from $\mathbf{y}^O$ and $\mathbf{y}^{BB}$ as follows. Bounding box parameters are read out from $\mathbf{y}^{BB}$ at locations of local maxima on a thresholded $\mathbf{y}^O$ (using a $3 \times 3$ nonmaxima suppression, NMS). The bounding boxes are refined by feeding them as prompts into a SAM decoder [12] on the already computed backbone features $\mathbf{f}^I$. The boxes are refitted to the masks by min-max operation and finally non-maxima suppression with IoU $= 0.5$ is applied to remove duplicate detections. This process thus yields the predicted bounding boxes $\mathbf{B}^P$ and their corresponding masks $\mathbf{M}^P$.

## 3.4 A novel loss for dense detection training

GeCotraining requires supervision on the dense objectness scores $\mathbf{y}^O$ and the bounding box parameters $\mathbf{y}^{BB}$. Ideally, a network should learn to predict points on objects that can be reliably detected by a NMS, and also from which the bounding box parameters can be reliably predicted. We thus propose a new dense object detection loss that pursues this property.

Following the detection step (Section 3.2) in the forward pass, a set of local maxima $\{i\}_{i=1:N_{\mathrm{DET}}}$ is identified by applying a NMS on $\mathbf{y}^O$ and keeping all maxima higher than the median response, to ensure detection redundancy. The maxima are then labelled as *true positives* (TP) and *false positives* (FP) by applying Hungarian matching [13] between their bounding box parameters $\{\mathbf{y}_i^{BB}\}_{i=1:N_{\mathrm{DET}}}$ and the ground truth bounding boxes $\{\mathbf{B}_j^{GT}\}_{j=1:N_{\mathrm{GT}}}$. To account for missed detections, centers of

the non-matched ground truth bounding boxes are added to the list of local maxima and labeled as *false negatives* (FN). The new training loss is thus defined as

$$\mathcal{L} = - \sum_{i \in \text{TP}} \text{gIoU}(\mathbf{y}_i^{BB}, \boldsymbol{B}_{\text{HUN}(i)}^{GT}) + \sum_{i \in \text{TP} \cup \text{FN}} (\mathbf{y}_i^{O} - 1)^2 + \sum_{i \in \text{FP}} (\mathbf{y}_i^{O} - 0)^2, \qquad (3)$$

where $\text{gIoU}(\cdot, \cdot)$ is the generalized IoU [25], and $\text{HUN}(i)$ is the ground truth index matched with the $i$-th predicted bounding box. Note that the new loss simultaneously optimizes the bounding box prediction quality, promotes locations with better box prediction capacity that can be easily detected by a NMS, and enables automatic hard-negative mining in the objectness score via FP identification.

## 4 Experiments

**Implementation details.**

Using the SAM [12] backbone, GeCo reduces the input image by a factor $r = 16$, and projects the features into $d = 256$ channels (Section 3). In DQE (Section 3.1), $N_P = 3$ iterations are applied in prototype generalization (1) and $N_Q = 2$ iterations in dense object query construction (2). Following the established test-time practice [20; 19; 26], the input image is scaled to fit $W_0 = H_0 = 1536$ if the average of the exemplars widths and heights is below 25 pixels, otherwise it is downscaled to fit the average of the exemplar width and height to 80 pixels and zero-padded to $W_0 = H_0 = 1024$. As in [20], the zero-shot GeCo is run twice, first to estimate the objects size and then again on the resized image.

**Training details.** With the SAM backbone frozen, GeCo is pretrained with the classical loss [20] for initialization and is then trained for 200 epochs with the proposed dense detection loss (3) using a mini-batch size of 8, AdamW [16] optimizer, with initial learning rate set to $10^{-4}$, and weight decay of $10^{-4}$. The training is done on 2 A100s GPUs with standard scale augmentation [20; 4] and zero-padding images to $1024 \times 1024$ resolution. For the zero-shot setup, the few-shot GeCo is frozen and only the zero-shot prototype extension is trained for 10 epochs. Thus *the same trained network* is used in all low-shot setups.

**Evaluation metrics and datasets.** Standard datasets are used. The FSCD147 [19] is a detection-oriented extension of the FSC147 [24], which contains 6135 images of 147 object classes, split into 3659 training, 1286 validation, and 1190 test images. The splits are disjoint such that target object categories in test set are not observed in training. The objects are manually annotated by bounding boxes in the test set [19], while in the train set, the bounding boxes are obtained from point estimates by SAM [35]. For each image, three exemplars are provided. The second dataset is FSCD-LVIS [19], derived from LVIS [8] and contains 377 categories. Specifically, the unseen-split is used (3959 training and 2242 test images), which ensures that test-time object categories are not observed during training.

The standard evaluation protocol [24; 26; 32] with Mean Absolute Error (MAE) and Root Mean Squared Error (RMSE) is followed to evaluate the counting accuracy. Following [19], Average Precision (AP) and Average Precision at IoU=50 (AP50) is used on the same output to evaluate the detection accuracy.

### 4.1 Experimental Results

**Few-shot counting and detection.** GeCo is compared with state-of-the-art density-based counters (which only estimate the total count) LOCA [4], CounTR [14], SAFECount [32], BMNet+ [26], VCN [22], CFOCNet [31], MAML [7], FamNet [24] and CFOCNet [31], and with detection-based counters C-DETR [19], SAM-C [18], PSECO [35], and DAVE [20], which also provide object locations by bounding boxes. Results are summarized in Table 1.

GeCo outperforms both recent state-of-the-art detection-based counters DAVE [20] and PSECO [35] by a 24% and 39% MAE, and a remarkable 27% and 51% RMSE on the test split, setting a new state-of-the-art in detection-based counting. Notably, GeCo outperforms all single-stage density-based counters (top part of Table 1) by a large margin, which makes it the first detection-based counter that outperforms the longstanding total count estimation winner LOCA [4] by a remarkable 27% MAE and 4% RMSE on test split. In this respect, GeCo closes the performance gap that has been present for several years between state-of-the-art density-, and detection-based counters.

Table 1: Few-shot density-based methods (top part) and detection-based methods (bottom part) performances on the FSCD147 [19].

| | Validation set | | | | Test set | | | |
|---|---|---|---|---|---|---|---|---|
| Method | MAE (↓) | RMSE(↓) | AP(↑) | AP50(↑) | MAE(↓) | RMSE(↓) | AP(↑) | AP50(↑) |
| GMN [17] ACCV18 | 29.66 | 89.81 | - | - | 26.52 | 124.57 | - | - |
| MAML [7] ICML17 | 25.54 | 79.44 | - | - | 24.90 | 112.68 | - | - |
| FamNet [24] CVPR21 | 23.75 | 69.07 | - | - | 22.08 | 99.54 | - | - |
| CFOCNet [31] WACV21 | 21.19 | 61.41 | - | - | 22.10 | 112.71 | - | - |
| BMNet+ [26] CVPR22 | 15.74 | 58.53 | - | - | 14.62 | 91.83 | - | - |
| VCN [22] CVPRW22 | 19.38 | 60.15 | - | - | 18.17 | 95.60 | - | - |
| SAFEC [32] WACV23 | 15.28 | 47.20 | - | - | 14.32 | 85.54 | - | - |
| CounTR [14] BMVC22 | 13.13 | 49.83 | - | - | 11.95 | 91.23 | - | - |
| LOCA [4] ICCV23 | 10.24 | 32.56 | - | - | 10.79 | 56.97 | - | - |
| C-DETR [19] ECCV22 | 20.38 | 82.45 | 17.27 | 41.90 | 16.79 | 123.56 | 22.66 | 50.57 |
| SAM-C [18] arXiv23 | 31.20 | 100.83 | 20.08 | 39.02 | 27.97 | 131.24 | 27.99③ | 49.17 |
| PSECO [35] CVPR24 | 15.31③ | 68.36③ | 32.12① | 60.02③ | 13.05③ | 112.86③ | 42.98② | 73.33② |
| DAVE [20] CVPR24 | 9.75② | **40.30①** | 24.20③ | 61.08② | 10.45② | 74.51② | 26.81 | 62.82③ |
| GeCo (ours) | **9.52①** | 43.00② | **33.51①** | **62.51①** | **7.91①** | **54.28①** | **43.42①** | **75.06①** |

In terms of detection performance, GeCo surpasses all state-of-the-art methods, including PSECO [35] which uses both, SAM [12] and CLIP [21] backbones, by 1% AP, and 2% AP50. Note that GeCo also outperforms PSECO in count prediction by a large margin (∼40%), which is crucial, as an ideal detection counter should deliver both accurate total count prediction as well as feature good object localization. In addition, GeCo also outperforms SAM-C, which is a low-shot counting and detection extension of SAM by 70%/55% MAE/AP. To demonstrate the impact of the refinement step in existing methods, we modified DAVE [20] by feeding predicted bounding boxes to SAM [12] as prompts, which results in a GeCo-like box refinement. Compared to modified DAVE, GeCo achieves 21% and 16% higher AP and AP50, respectively, indicating that the reason for the excellent performance of GeCo lies in its architecture, rather than in segmentation-based refinement.

Figure 3 visualizes detections for qualitative analysis[1]. GeCo predicts bounding boxes of superior quality for elongated objects (row 1), validating the selection of bounding box prediction locations. On detecting complex, non-blob-like objects (row 2), GeCo outperforms concurrent methods, by more accurately generalizing the prototypes. In densely populated scenes (row 3), GeCo achieves higher accuracy in both count and bounding box predictions. In comparison with state-of-the-art, GeCo features better object discrimination (row 4), which can be attributed to better prototype generalization in DQE (Section 3.1) and hard negative mining in the new loss from Section 3.4.

We further evaluate GeCo on FSCD-LVIS [19]. Results in Table 2 show that GeCo outperforms the best method by significant 178% and 73% in AP and AP50, respectively, and performs on-par in terms of MAE. The experiment supports the results on FSCD147.

Table 2: Few-shot counting and detection on the FSCD-LVIS [19] "unseen" split.

| | Count | | Detection | |
|---|---|---|---|---|
| Method | MAE(↓) | RMSE(↓) | AP(↑) | AP50(↑) |
| FSDetView-PB [29] TPAMI22 | 28.99 | 40.08 | 1.03 | 2.89 |
| AttRPN-PB [6] CVPR22 | 39.16 | 46.09 | 3.15 | 7.87 |
| C-DETR [19] ECCV22 | 23.50③ | 35.89③ | 3.85③ | 11.28③ |
| DAVE [20] CVPR24 | 15.47② | **25.95①** | 4.12② | 14.16② |
| GeCo (ours) | **15.26①** | 28.80② | **11.47①** | **24.49①** |

**One-shot counting and detection.** In the one-shot counting setup, a single exemplar is considered. Table 3 shows comparison with the recent density- and detection-based methods. GeCo outperforms all state-of-the-art single-stage density-based counters, outperforming $LOCA_{1-shot}$ [4] version specifically trained for the one-shot setup, by a significant margin of 35% MAE and 20% RMSE on validation and test split, respectively. GeCo also outperforms state-of-the-art method PSECO [35] by

---

[1] See supplementary material for more visualizations

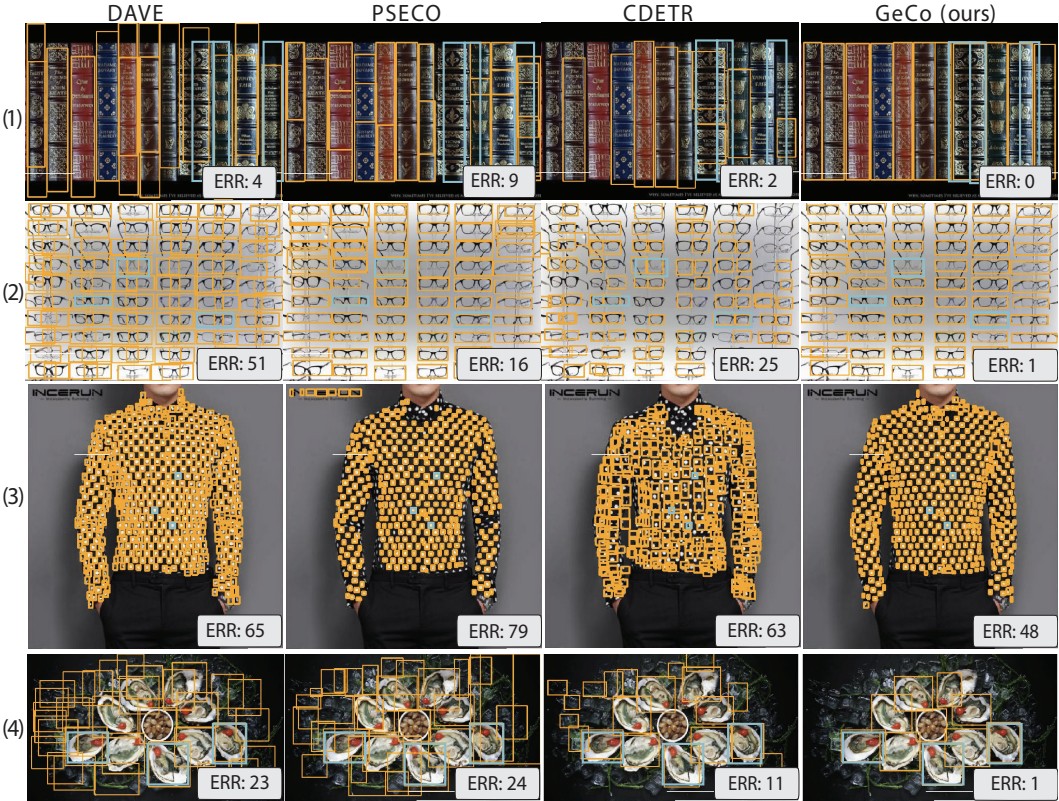

Figure 3: Compared with state-of-the-art few-shot detection-based counters DAVE [20], PSECO [35], and C-DETR [19], GeCo delivers more accurate detections with less false positives and better global counts. Exemplars are delineated with blue color, while segmentations are not shown for clarity.

4% AP and 5% AP50, and by significant 45% MAE and 49% RMSE on test split. These results show that GeCo features remarkable robustness to the number of exemplars since a single network (without re-training or fine-tuning) is used in both three- and one-shot setups. In particular, the performance drops by only 2%/11% of MAE/RMSE and 1%/1% AP/AP50 on the test split between both setups. In a *one-shot* setting, GeCo surpasses state-of-the-art *three-shot* models. Specifically, one-shot GeCo achieves 22% and 20% lower MAE and RMSE, respectively, compared to three-shot DAVE, and outperforms three-shot PSECO by 38% and 46% on the FSCD147 test set. These results highlight the robustness of GeCo to the number of exemplars, demonstrating its ability to handle inputs with lowered visual diversity.

**Zero-shot counting and detection.** Table 4 reports the results of the zero-shot GeCo compared with best zero-shot variants of the density-based counters, LOCA [4], CounTR [14], RepRPN-C [23], RCC [10] and with the zero-shot variant of the best detection-based counter DAVE [20]. GeCo outperforms DAVE [20] by a significant margin of 14% MAE and 6% RMSE on the test set. Furthermore, it outperforms all density-based methods and sets a new state-of-the-art result on FSC [24] benchmark, by outperforming the top-performer CounTR [14] by impressive 6% MAE on the test set. Since the zero-shot variant of the recent detection-based counter PSECO [35] does not exist, we include its prompt-based variant for complete evaluation (i.e., target object class is specified by a text prompt). Even in this setup, the zero-shot GeCo outperforms the prompt-based PSECO by 20% MAE 16% RMSE, and 2% AP50 demonstrating great robustness to different counting and detection scenarios.

**Mutliclass images.** To further verify the robustness of the proposed method, we validate it on a subset of FSCD147, that contain images with multiple object classes (FSCD147$_{mul}$) [20]. Results in Table 5 indicate that most state-of-the-art methods non-discriminatively count all objects in an image due to prototype over-generalization. GeCo outperforms all single-stage density-, and detection-based counters on multiclass images by at least 60%/67% in MAE/RMSE. This further verifies the

Table 3: One-shot density-based methods (top) and detection-based methods (bottom) on the FSCD147 [19].

| Method | Validation set | | | | Test set | | | |
|---|---|---|---|---|---|---|---|---|
| | MAE (↓) | RMSE(↓) | AP(↑) | AP50(↑) | MAE(↓) | RMSE(↓) | AP(↑) | AP50(↑) |
| GMN [17] ACCV18 | 29.66 | 89.81 | - | - | 26.52 | 124.57 | - | - |
| CFOCNet [31] WACV21 | 27.82 | 71.99 | - | - | 28.60 | 123.96 | - | - |
| FamNet [24] CVPR21 | 26.55 | 77.01 | - | - | 26.76 | 110.95 | - | - |
| BMNet+ [26] CVPR22 | 17.89 | 61.12 | - | - | 16.89 | 96.65 | - | - |
| CounTR [14] BMVC22 | 13.15 | 49.72 | - | - | 12.06 | 90.01 | - | - |
| LOCA$_{\text{1-shot}}$ [4] ICCV23 | 11.36 | 38.04 | - | - | 12.53 | 75.32 | - | - |
| PSECO [35] CVPR24 | 18.31③ | 80.73③ | 31.47② | 58.53② | 14.86③ | 118.64③ | 41.63② | 70.87② |
| DAVE$_{\text{1-shot}}$ [20] CVPR24 | 10.98② | 43.26② | 18.00③ | 52.37③ | 11.54② | 86.62② | 19.46③ | 55.27③ |
| GeCo (ours) | **9.97**① | **37.85**① | **32.82**① | **61.31**① | **8.10**① | **60.16**① | **43.11**① | **74.31**① |

Table 4: Zero-shot density-based methods (top part), and detection-based methods (bottom part) on the FSCD147 [19]. The symbol ∗ denotes methods that also use text prompts as input.

| Method | Validation set | | | | Test set | | | |
|---|---|---|---|---|---|---|---|---|
| | MAE (↓) | RMSE(↓) | AP(↑) | AP50(↑) | MAE(↓) | RMSE(↓) | AP(↑) | AP50(↑) |
| RepRPN-C [23] ACCV22 | 29.24 | 98.11 | - | - | 26.66 | 129.11 | - | - |
| RCC [10] arXiv22 | 17.49 | 58.81 | - | - | 17.12 | 104.5 | - | - |
| CounTR [14] BMVC22 | 17.40 | 70.33 | - | - | 14.12 | 108.01 | - | - |
| LOCA [4] ICCV23 | 17.43 | 54.96 | - | - | 16.22 | 103.96 | - | - |
| PSECO [35]∗ CVPR24 | 23.90③ | 100.33③ | - | - | 16.58③ | 129.77③ | 41.14② | 69.03② |
| DAVE [20] CVPR24 | 15.71② | **60.34**① | 16.31② | 46.87② | 15.51② | 116.54② | 18.55③ | 50.08③ |
| GeCo (ours) | **14.81**① | 64.95② | **31.04**① | **58.30**① | **13.30**① | **108.72**① | **41.27**① | **70.09**① |

robustness of the proposed architecture, which benefits from the hard-negative mining in the proposed loss function, leads to more discriminative prototype construction and false positive reduction.

## 4.2 Ablation study

**Dense object detection loss.** To analyze the contribution of the new dense detection loss from Section 3.4, we trained GeCo using the standard loss [20; 35] that forms the ground truth objectness score by placing unit Gaussians on object centers – this variant is denoted by GeCo$_{\text{Gauss}}$. Table 6 shows that this leads to a substantial drop in total count estimation (38% RMSE, and 34% MAE) as well as in object detection (6% AP, and 3% AP50). Qualitative results are provided in Figure 4. As observed in columns 3 and 5, the classical unit-Gaussian-based loss [20; 35] forces the network to predict object locations from the object centers, which are not necessarily optimal for bounding box prediction. In contrast, the proposed dense detection loss enables the network to learn optimal point prediction, which more accurately aggregates information of the object pose. Columns 1 and 2 indicate that the new loss leads to superior detection of objects composed of blob-like structures avoiding false detections on individual object parts. Furthermore, the hard-negative mining integrated in the new loss design leads to better discriminative power of the detections and subsequent reduction of false positives (column 4).

Table 5: Performance on FSCD147 [19] test split, and its multiclass subset FSCD147$_{\text{mul}}$.

| Method | FSCD147 | | FSCD147$_{\text{mul}}$ | |
|---|---|---|---|---|
| | MAE(↓) | RMSE(↓) | MAE(↓) | RMSE(↓) |
| C-DETR [19] ECCV22 | 16.79 | 123.56 | 23.09 | 30.09 |
| PSECO [35] CVPR24 | 13.05 | 112.86 | 25.73 | 44.95 |
| LOCA [4] ICCV23 | 10.79③ | 56.97② | 21.28 | 43.67 |
| CounTR [14] BMVC22 | 11.95 | 91.23 | 14.56③ | 27.41③ |
| DAVE [20] CVPR24 | 10.45② | 74.51③ | **3.09**① | **5.28**① |
| GeCo (ours) | **7.91**① | **54.28**① | 5.88② | 9.17② |

Table 6: Ablation study on the FSCD147 [19] validation split.

| Method | Counting | | Detection | |
|---|---|---|---|---|
| | MAE($\downarrow$) | RMSE($\downarrow$) | AP($\uparrow$) | AP50($\uparrow$) |
| GeCo | **9.52** | **43.00** | **33.51** | **62.51** |
| GeCo$_{\text{Gauss}}$ | 12.79 | 59.33 | 31.43 | 60.73 |
| GeCo$_{\overline{\text{HQ}}}$ | 10.04 | 47.11 | 33.08 | 62.50 |
| GeCo$_{\overline{\mathbf{p}^s}}$ | 9.97 | 46.93 | 32.56 | 61.19 |
| GeCo$_{\overline{\text{Ref}}}$ | 10.26 | 43.33 | 24.63 | 61.57 |
| GeCo$_{\overline{\mathbf{Q}}}$ | 10.32 | 45.14 | 33.01 | 61.68 |
| GeCo$_{\text{DETR}}$ | 11.45 | 52.46 | 32.24 | 61.60 |

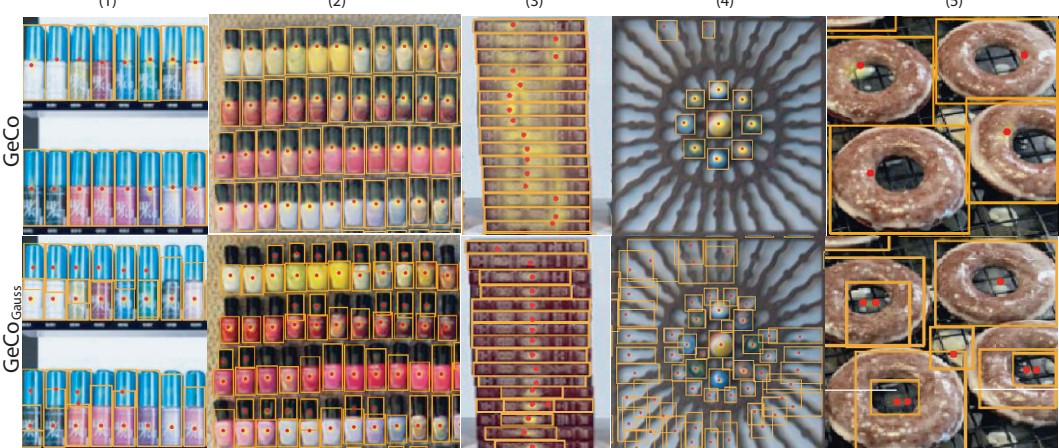

Figure 4: Response maps (in yellow), and locations for bounding box predictions (red dots) when using the proposed (first row) and the standard [20; 4; 35] (second row) training loss.

**Architecture.** To evaluate the impact of concatenating the SAM-HQ [11] features in the query *unpacking* process in the DQD module (Section 3.2), we remove these features in GeCo$_{\overline{\text{HQ}}}$. Table 6 shows a counting performance drop 5% MAE and 10% RMSE. To validate the importance of modeling exemplar shapes, i.e., width and height, with prototypes $\mathbf{p}^S$, we omit them in GeCo$_{\overline{\mathbf{p}^S}}$. We observe a substantial performance decrease of 5% MAE, and 9% RMSE. Finally, we remove bounding box refinement in the detection refinement module (Section 3.3), and denote the variant as GeCo$_{\overline{\text{Ref}}}$. While this does not affect the global count estimation accuracy, we observe a 26% decrease in AP and 2% decrease in AP50. It is worth noting, that bounding box refinement improves the accuracy of predicted bounding boxes, however it does not enhance object presence detection.

To verify the importance of the DQE module (Section 3.1), we replace the dense object queries $\mathbf{Q}$ construction step (2) with a standard self-attention, i.e., $\mathbf{Q} = \text{SA}(\mathbf{P})_{3\times}$. This leads to a 8% MAE and 5% RMSE performance drop, verifying the proposed approach. To evaluate the importance of using image features as queries in (2), we change the object query construction to $\mathbf{Q}_j = \text{CA}(\text{SA}(\mathbf{Q}_{j-1}), \mathbf{f}^I, \mathbf{f}^I)$ to follow a standard DETR [1]-like approach, and denote it as GeCo$_{\text{DETR}}$. We observe a 20% MAE and 22% RMSE decrease in counting performance.

## 5 Conclusion

We proposed GeCo, a novel single-stage low-shot counter that integrates accurate detection, segmentation, and count prediction within a unified architecture, and covers all low-shot scenarios with a single trained model. GeCo features remarkables dense object query formulation, and prototype generalization across the image, rather than just into a few prototypes. It employs a novel loss function specifically designed for detection tasks, avoiding the biases of traditional Gaussian-based losses. The loss optimizes detection accuracy directly, leading to more precise detection and counting.

The main limitation of the presented method is that it cannot process arbitrarily large images, due to memory constraints, since it, as all current methods, operates globally. In future work, we will explore local counting, incremental image-wide count aggregation, optimizing inference speed utilizing a faster backbone [34].

Extensive analysis showcases that GeCo surpasses the best detection-based counters by approximately 25% in total count MAE, achieving state-of-the-art performance in a few-shot counting setup and demonstrating superior detection capabilities. GeCo showcases remarkable robustness to the number of provided exemplars, and sets a new state-of-the-art in one-shot as well as zero-shot counting.

**Acknowledgements.** This work was supported by Slovenian research agency program P2-0214 and projects J2-2506, L2-3169, Z2-4459, J2-60054, and by supercomputing network SLING (ARNES, EuroHPC Vega - IZUM).

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

# A  Supplemental material

This supplementary material provides additional comparisons of GeCo with state-of-the-art under a non-standard experiment, and provides additional qualitative examples.

**Performance analysis on a non-standard experiment**. The analysis of the detection methods in Section 4 adheres to the standard evaluation protocol [19; 20], where a method predicts a set of bounding boxes for each image. The estimated count is the total number of predicted bounding boxes, and evaluated by the MAE/RMSE measures, while the detection accuracy is evaluated by AP/AP50 measures. Both measures are computed on *the same set* of output bounding boxes.

However, in the PSECO [35] paper, the reported evaluation deviated from the standard one in an important detail. Namely *different* outputs were evaluated under MAE/RMSE and AP/AP50 to fully evaluate the different properties of the method. AP/AP50 was computed in *all* output bounding boxes, while the MAE/RMSE were computed on a subset of the boxes, obtained by thresholding the response score. In Section 4, we evaluated all methods, including PSECO under the standard experiment. Nevertheless, we additionally report GeCo evaluated under the said non-standard PSECO experiment in Table 7.

Even in this setup, GeCo outperforms PSECO by 4%/4% AP/AP50, and 1%/2% AP/AP50, on validation and test set, respectively, again with a substantially lower global count errors (~50% MSE/RMSE reduction). These results shed an important insight. A method producing false positives, which increase the count errors and reduce its usefulness for counting, might achieve good detection-oriented performance measures. Thus for counting performance evaluation, the MAE/RMSE should be considered primary measures, while AP/AP50 should be secondary, as they are less strict towards false positive detections.

Table 7:  Few-shot detection-based counting evaluation on FSCD147 [19] under the non-standard evaluation protocol [35].

| Method | Validation set | | | | Test set | | | |
| --- | --- | --- | --- | --- | --- | --- | --- | --- |
| | MAE ($\downarrow$) | RMSE($\downarrow$) | AP($\uparrow$) | AP50($\uparrow$) | MAE($\downarrow$) | RMSE($\downarrow$) | AP($\uparrow$) | AP50($\uparrow$) |
| PSECO [35] | 15.31② | 68.34② | 32.71② | 62.03② | 13.05② | 112.86② | 43.53② | 74.64② |
| GeCo (ours) | **9.52**① | **43.00**① | **34.07**① | **64.23**① | **7.91**① | **54.28**① | **43.89**① | **76.18**① |

**Performance in crowded scenes**. To evaluate counting performance in crowded scenes, we constructed a subset of the FSCD147 test set by including images with at least 200 objects and a maximal average exemplar size of 30 pixels. Notably, the new subset contains 42 images, averaging 500 objects per image, thus featuring dense scenes with small objects. Three top-performing methods from Table 1 were included in the comparison and are shown in Table 8. GeCo outperforms both PSECO and DAVE by a significant margin, e.g., outperforming DAVE by 23% in MAE and 36% in RMSE, which demonstrates superior counting performance on small, densely populated objects.

Table 8:  Few-shot counting in crowded scenes, comparing the top-three detection-based counters from Table 1.

| | MAE | RMSE |
| --- | --- | --- |
| PSECO [35] CVPR24 | 173.64 | 594.91 |
| DAVE [20] CVPR24 | 81.38 | 383.93 |
| GeCo (ours) | 62.60 | 242.82 |

**Qualitative results.** Figure 5 compares GeCo with PSECO [35], which achieves the best AP/AP50 measures among the related counters. GeCo shows robust performance, achieving high precision (see Figure 5 block 1), while achieving high recall (see Figure 5 block 2). This is challenging for related methods, particularly in densely populated scenes or with small objects. Furthermore, GeCo outperforms PSECO on elongated or more complex objects (see Figure 5 block 3), better exploiting the exemplars.

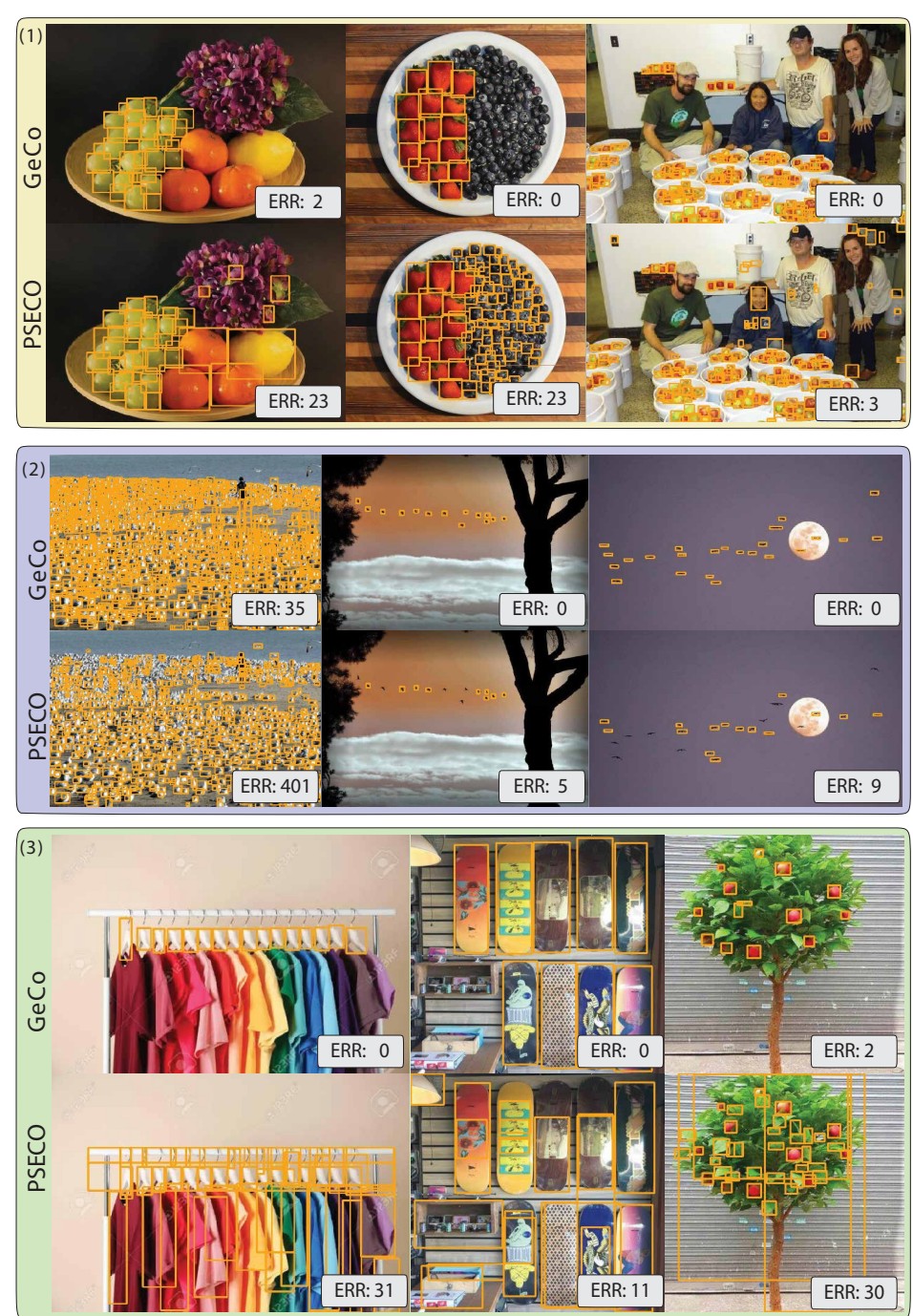

Figure 5: Comparison of few-shot counting on FSCD147. Exemplars are shown with red color and ERR indicates count error.

Figure 6 visualizes the segmentations produced by GeCo, in a few-shot setup, of various objects in diverse scenes. GeCo is robust to noise, achieves discriminative segmentations, and performs well on elongated, non-blob-like objects and in dense scenarios. Figure 7 compares GeCo with all state-of-the-art detection counters [20; 19; 35]. GeCo achieves superior counting performance, and predicts more accurate bounding boxes.

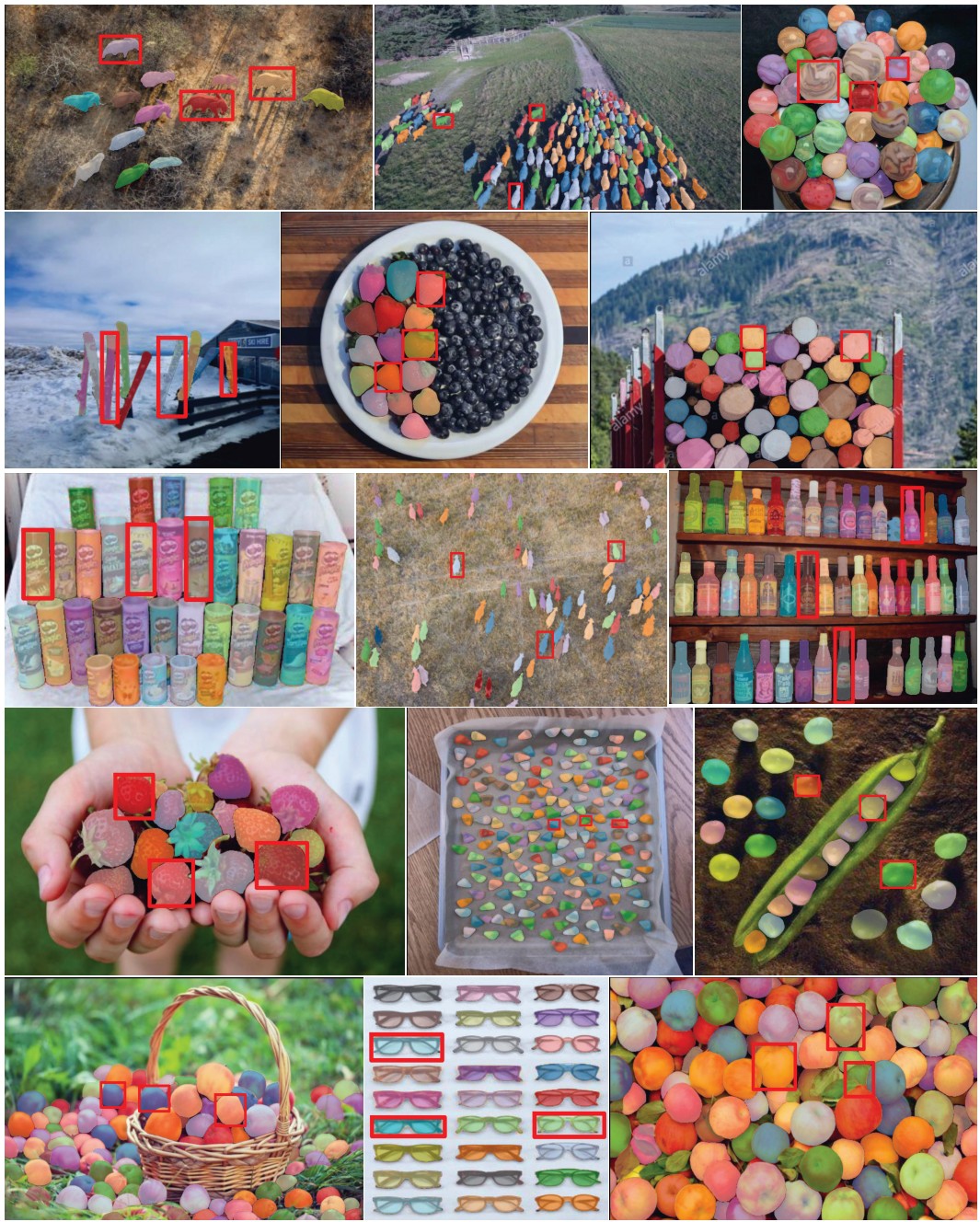

Figure 6: Segmentation quality of GeCo on diverse set of scenes and object types. Exemplars are denoted by red bounding boxes.

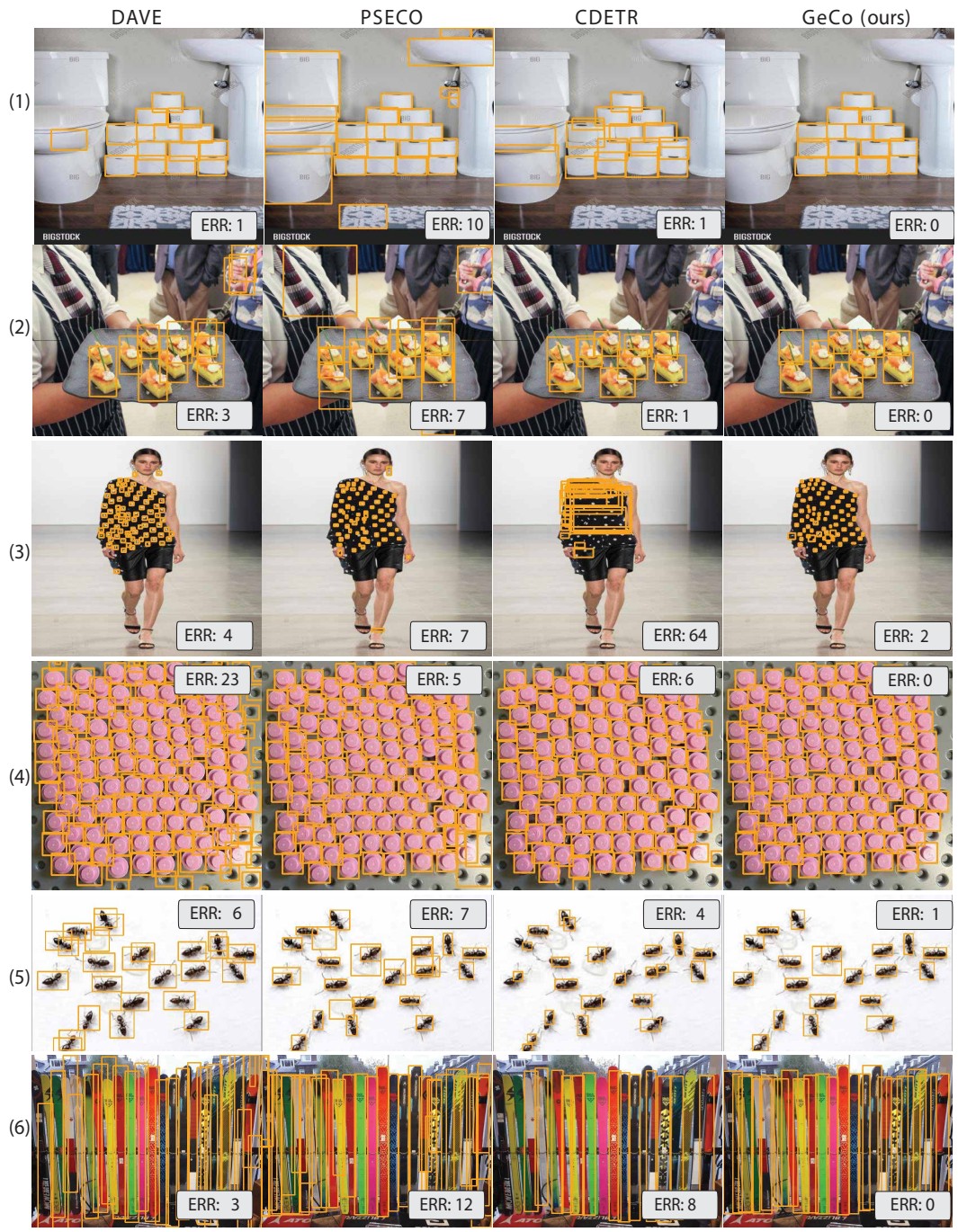

Figure 7: Comparison of few-shot counting and detection on FSCD147. ERR indicates count error.

In Figure 8 performance of GeCo is qualitatively demonstrated on examples with high intra-class variance. Image (a) displays marbles of various colors and textures (notable visual intra-class variance), all correctly detected and still distinguished from a visually similar coin. Example (b) shows donuts with different colors of decorations, all accurately counted and detected by GeCo. Image (c) contains bottles of various sizes, shapes, and colors, each with a distinct sticker. Image (d) features transparent food containers with differently colored and shaped fruits inside, successfully detected despite significant visual diversity. Examples (e) and (f) illustrate GeCo's robustness in detecting objects with high shape variance, including partially visible birds (notable object shape intra-class variance).

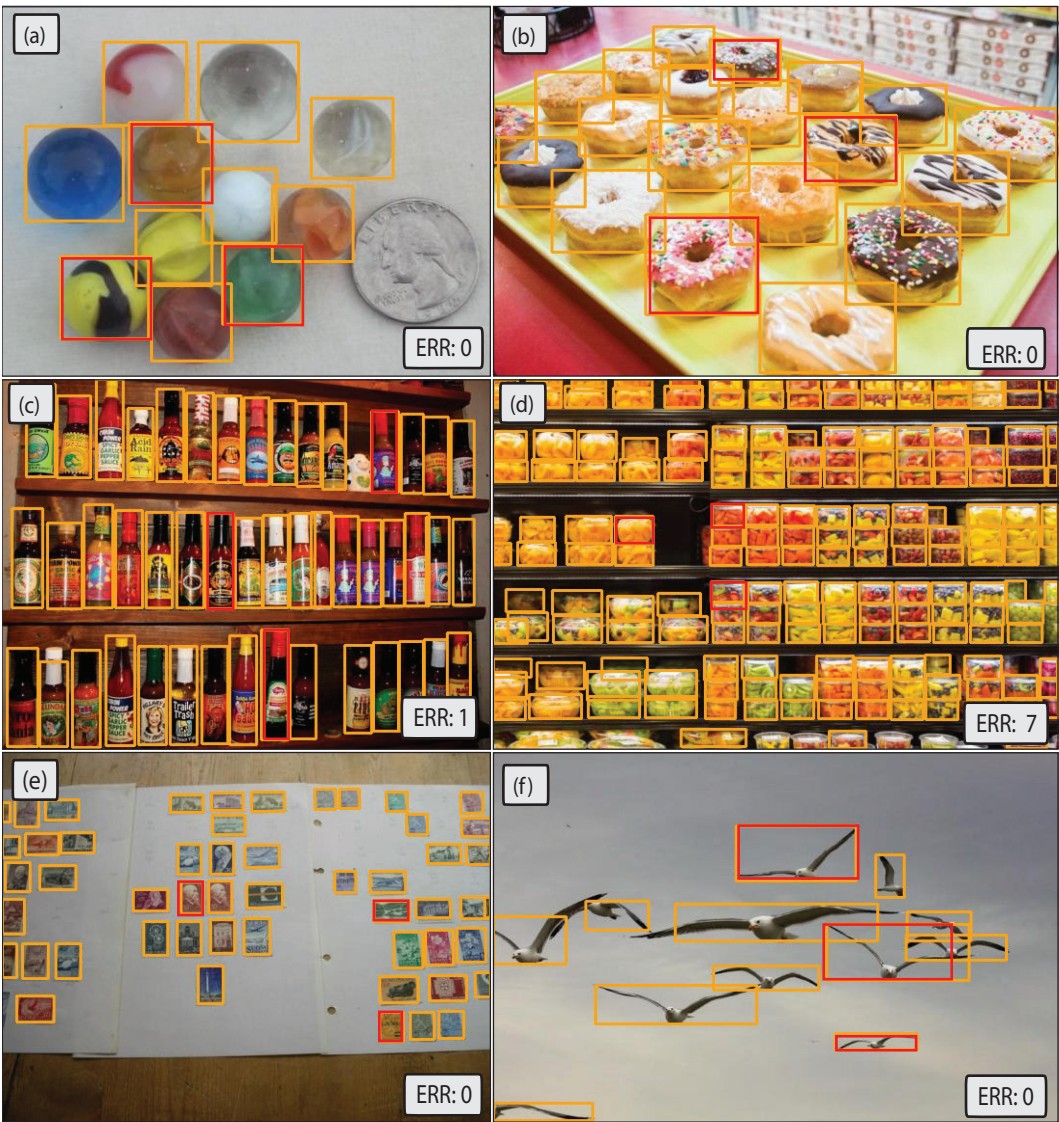

Figure 8: Few-shot detection and counting with GeCo on images with high intra-class object appearance variation. Orange and red bounding boxes denote detections and exemplars, respectively. Count error is denoted by ERR.

