# OpenReview forum: "A Novel Unified Architecture for Low-Shot Counting by Detection and Segmentation"
_NeurIPS.cc/2024/Conference — NeurIPS 2024 poster_

### Official Review · Reviewer_QayL · 2024-07-11

**Soundness:** 3
**Presentation:** 3
**Contribution:** 2
**Rating:** 6
**Confidence:** 4

**Summary:**

This paper introduces GeCo, a novel unified architecture for low-shot counting that integrates object detection and segmentation. GeCo addresses the limitations of current state-of-the-art methods by generalizing object prototypes across diverse appearances and introducing a new counting loss that directly optimizes the detection task. The architecture achieves significant improvements in detection accuracy and count estimation over previous methods and sets a new benchmark in low-shot counting.

**Strengths:**

1. GeCo combines object detection and segmentation in a single framework, enhancing efficiency and performance.
2. The novel dense object query formulation improves the generalization of object prototypes across diverse appearances, reducing false positives.
3. The proposed counting loss directly optimizes detection tasks, leading to more accurate object counts and better handling of annotation noise.
4. GeCo surpasses existing few-shot detection-based counters by approximately 25% in total count MAE and shows superior detection accuracy. The method demonstrates substantial robustness to the number of exemplars, performing well even in one-shot and zero-shot scenarios.

**Weaknesses:**

1. While SAM's pre-training provides strong generalization with other potential backbone networks, it is unclear if similar results could be achieved with other networks like ResNet or EfficientNet. Thus the complexity and computational requirements of the proposed GeCo architecture might limit its scalability and real-time applicability, especially in resource-constrained environments.
2. Although the method is claimed to be low-shot friendly, the paper does not provide explicit architectural components or techniques specifically designed for low-shot scenarios.
3. The performance of GeCo on highly heterogeneous datasets, where object appearances vary significantly within the same category, is not adequately demonstrated.
4. The effectiveness of this method in scenarios involving small objects or high-density areas, such as crowd density estimation or face detection in dense crowds, has not been well explored as this type of dataset has large-scale variations.

**Questions:**

1. Why was the SAM framework specifically chosen over other backbone networks? How does it compare in terms of generalization and performance with other pre-trained models like ResNet or EfficientNet?
2. What specific design elements make GeCo particularly suitable for low-shot learning? Are there any modifications or components that specifically address the challenges of low-shot scenarios?
3. How does GeCo perform on datasets with significant intra-class variability? Can it generalize well to objects with a high degree of appearance variation within the same category?

**Limitations:**

1. The method's performance still heavily depends on the quality and representativeness of the provided exemplars. Poor quality or unrepresentative exemplars could negatively impact the detection and counting accuracy.
2. While the dense object query formulation improves generalization, the method's ability to handle extreme variability in object appearances within the same category needs further validation.
3. The effectiveness of GeCo in scenarios involving small objects or highly dense regions (e.g., crowd counting, face detection in crowds) is not thoroughly evaluated. These scenarios often present unique challenges that may require additional considerations.

---

> ### Author Rebuttal · Authors · 2024-08-06
>
> >**Why was SAM backbone chosen over other backbones?**
>
> We primarily wanted to use SAM as a box refiner, due to its accurate mask prediction, which can easily be converted to a bounding box. For computational efficiency and to keep the framework unified, we then decided to also use the SAM backbone for feature extraction in GeCo. This makes GeCo simply another head on the SAM backbone, thus adding another capability to the SAM Swiss-knife.
> However since the proposed GeCo detection pipeline is general, any backbone network could be used instead of SAM in the image encoding stage. We do not expect significant performance drop due to a different backbone, since the networks such as Resnet have been shown to produce features with excellent detection capabilities.
>
> Note, however, that, to optimize the inference speed and memory consumption of GeCO, SAM could be replaced with the recently proposed Fast Segment anything [1] that speeds up the original SAM inference 50x and features only a marginal segmentation accuracy loss. This will be the subject of future work and we will add this discussion into the final paper.
>
> [1] Zhao, X., et. al (2023). Fast Segment Anything. arXiv [Cs.CV].
>
> >**Why is GeCo particularly suitable for low-shot scenarios?**
>
> The proposed prototype construction addresses a specific low-shot prototype learning issue in counters. The current state-of-the-art low-shot counters predominantly construct a few prototypes from the annotated exemplars. Since the few prototypes are expected to cover a broad range of object appearances, the trained networks typically learn to over-generalize which leads to false positive detections. In contrast, GeCo non-parametrically (i.e., exploits all feature locations) constructs dense location-specific prototypes, thus avoiding the need for only few prototypes to cover the entire appearance diversity of the present objects. This ensures a high recall at high precision (i.e., minimal false positive detection) as verified in our experiments. Furthermore, the non-parametric formulation enables excellent performance in highly dense and crowded regions, where classical DETR-like counters, which can handle only a limited number of queries, fail.
>
> Another low-shot counting modification is the new detection-optimized loss, which avoids the standard losses that force the network to predict a unit Gaussian on the object center and are prone to center annotation noise issues and ad-hoc setting of the kernel width hyperparameter. The new loss avoids these issues completely by training the network to predict response maps on which objects are reliably detected by nms. Results show significant performance boosts.
>
> >**How does GeCo perform on high intra-class variability?**
>
> We report in the paper the GeCo performance on the FSCD-LVIS dataset, which is derived from a detection dataset and in which the objects feature a high intra-class variability. The results are reported in Table 2 of the paper. Note that  GeCo outperforms all state-of-the-art by a significant margin, implying superior capability in handling the visual diversity.
> For qualitative analysis, we show examples of objects with significant visual intra-class variability in Figure 1 (see attached rebuttal PDF) which will be added to the supplementary material in the final version of the paper.
>
> >**Performance depends on the quality of exemplars.**
>
> We agree that exemplar annotation quality plays an important role, and that all low-shot counting algorithms rely on the accuracy and representativeness of the exemplars. Since only a few exemplars (~3) are required, it is safe to assume that in practice the user will pay attention to their annotation.
>
> Nevertheless, we did include the experiment where the number of exemplars varied from 3 to 1 (Section 4.1, Table 1 and 2). Performance of GeCo drops by only 2\%/11\% of MAE/RMSE and 1\%/1\% AP/AP50 on the test split between three-shot and one-shot counting, which is almost a negligible performance drop, indicating a degree of robustness to exemplar selection.
>
> The results are even more impressive when comparing the **one-shot GeCo** to the **three-shot sota** methods. In particular, the one-shot GeCo outperforms the three-shot DAVE by 22\% and 20\%, and three-shot PSECO by 38\% and 46\% in terms of MAE and RMSE on the test set of FSCD147, respectively. This is a strong indicator that GeCo does not require finely selected exemplars that cover a broad range of appearances, as might be the case with sota. We attribute this to dense prototype and query construction process in GeCo. We will more clearly expose this property in the final version of the paper and we plan to further explore this interesting aspect that the reviewer pointed out in our future work.
>
> >**What is performance with small objects or highly dense regions (eg. crowd counting)?**
>
>
> To address the reviewer’s concern, but still remain in the object-agnostic low-shot counting and detection setup, we created a subset of the FSCD147 test set, where only images with at least 200 objects with an average exemplar size of  30 pixels are considered. This subset consists of 42 images (~500 objects per image in average), making it a solid representative of dense scenes with small objects. We evaluated the three top-performing methods from Table 1 (including GeCo) on this dataset. Results are shown in the table below. GeCo outperforms both PSECO and DAVE, with the latter being outperformed by 23% in MAE and a significant 36% in RMSE.  We will add this table to the supplementary material of the final paper. Note that performance comparison in dense scenes is already qualitatively verified in an example in the supplementary material, Figure 5, example (2).
>
> |       | MAE    | RMSE   |
> | ----- | ------ | ------ |
> | PSECO | 173.64 | 594.91 |
> | DAVE  | 81.38  | 383.93 |
> | GeCo  | 62.60  | 242.82 |

---

> > ### Comment · Reviewer_QayL · 2024-08-12
> >
> > Thank you for the author's response, which solved most of my problems.
> >
> > SAM may give GeCo some benefit by looking at a lot of different data, which isn't fair to other comparison methods. The author said that any backbone network can be used instead of SAM in the image encoding stage, but it is not clear whether the performance will be significantly reduced because of the different backbones.
> >
> > Therefore I maintain my original rating:  6 Weak Accept

---

> > > ### Author Response · Authors · 2024-08-13
> > >
> > > We thank the reviewer for the positive comment and we are happy to clarify the remaining point about the SAM backbone and comparison fairness. Note that recent state-of-the-art (PSECO) also uses SAM as a backbone, but GeCo outperforms it substantially, thus comparison seems fair. To further clarify, we replaced SAM backbone in GeCo with the classical ResNet. As reviewer anticipated, the performance drops a bit, but remains high, i.e., MAE (7.91-> 9.01) RMSE (54.28 -> 48.01), and this GeCo version still outperforms all state-of-the-art, including PSECO that uses the SAM backbone. We conclude that the SAM backbone is not the main contributor to good counting performance of GeCo.

---

> > > > ### Comment · Reviewer_QayL · 2024-08-14
> > > > **Reply**
> > > >
> > > > Thank you. My recommendation is 6-Weak Accept

---

### Official Review · Reviewer_HCSh · 2024-07-13

**Soundness:** 3
**Presentation:** 2
**Contribution:** 2
**Rating:** 5
**Confidence:** 4

**Summary:**

Paper tackles the task of few-shot and zero-shot class agnostic counting, and presents Geco, a unified counting framework that can detect, segment and count objects. Geco uses SAM backbone for feature extraction, and implements Dense Query Encoder (DQE) and Dense Query Decoder (DQD) to detect prototypes. For few shot setup, prototypes are obtained from the exemplars, and zero shot setup uses learned objectness prototypes. DQD outputs a set of dense detections, which are further refined by the SAM decoder, by considering DQD detections as prompts for the SAM decoder.

**Strengths:**

Paper presents a unified approach for few shot and zero shot class agnostic object counting which achieves good results on standard benchmark datasets.

**Weaknesses:**

1. Lines 110 - 114: authors claim that existing low shot counting approaches uses a small number of prototypes or exemplars, often leading to false detections. Authors say their approach is different from these existing approaches. However, eqn 1 in Sec 3.1 appears to be conceptually similar to the existing approaches, i.e. few prototypes p are used to create keys and values in a cross attention layer, where the query is formed by the image features.
2. Looking at the ablation study in table 6, it appears that removing SAM decoder degrades the performance of Geco significantly in terms of AP, which raises some questions about the effectiveness of the approach. How does Geco perform without SAM decoder on the test set of FSCD147 ?
3. Related to my point 2 above and the ablation study in table 2: it would be useful to have a baseline without using Geco, by simply using the exemplars as prompts for the SAM decoder.  This baseline does not require any new training.

**Questions:**

I would request the authors to kindly address the points I raised in Weaknesses section.

**Limitations:**

Yes.

---

> ### Author Rebuttal · Authors · 2024-08-06
>
> >**Prototype construction is conceptually similar to related methods.**
>
> While Eq. 1 is indeed conceptually similar to the existing methods, there are fundamental differences in the function of the output and subsequent steps (Eq. 2).
>
> The existing methods perform detection by correlating exemplar prototypes directly with the image features and decoding the resulting features into a density map, or applying DETR-like detection on them. In both cases the features are assumed to have desired objects “highlighted” and others “suppressed”. Since the prototypes are expected to cover a broad range of appearances, the trained networks typically over-generalizes to achieve a high recall which leads to false positive detections.
>
> Concretely, related methods (e.g., LOCA) transfer information from the image into prototypes (i.e., attention keys and values are image features, while prototypes are attention queries) to enrich the prototypes, which leads to prototype over-generalization, since all appearance diversity is packed into only three prototypes, which are ultimately correlated with the features to detect object locations.
>
> A related method, CounTR, uses image features as attention queries and prototypes as attention keys and values, which is indeed similar to GeCo. But the result of this operation are “prototype-matched” features with objects “highlighted”, which are then decoded into a density map. This is functionally fundamentally different from GeCo. Since the information of only three prototypes is transferred into each image feature, the few prototypes are thus required to cover all appearance variations of target objects in the image, often, leading to over-generalization.
>
> In contrast, in GeCo, Eq. 1 transforms the few prototypes (Roi-pooled visual exemplars, and shapes exemplars) into a dense set of location-specific prototypes $\mathbf{P}$, i.e., one prototype per image location (feature space resolution). These prototypes are then not decoded into density map as in related works, but are used to construct dense local object queries $\mathbf{Q}$ through further attention mechanisms (Eq. 2). The latter are finally decoded into object detections. This process avoids over-generalization and false detections, and from this perspective differs from the existing methods. We will emphasize these points in the camera-ready.
>
> >**Performance decrease upon removing SAM decoder. Show ablation performance on FSCD147 test set as well.**
>
> The SAM decoder (refinement module) primarily just adjusts the predicted box boundaries, thus improving the **accuracy** of box predictions, but minimally changes the overall count (i.e., object detection), as is evident from the results in Table 6. In particular, the counting performance of GeCo$\_{\overline{\text{Ref}}}$ is nearly identical to the original GeCo (only 7% MAE and 1% RMSE performance drop). Similarly, the AP50, which indicates detection robustness, also remains consistent (a mere 2% drop). However, the AP of GeCo is higher (26%) since the detected boxes more accurately encompass the objects. We will add this discussion in the camera-ready.
>
> The reviewer mentions ablating GeCo on the test set. Ablation experiments are not performed on the test set in the paper, since the test set is meant for final performance evaluation only, by the FSCD protocol to prevent overfitting. However, to answer the reviewer’s question, we have run GeCo as requested: GeCo$\_{\overline{\text{Ref}}}$ achieves 8.96 MAE, 51.11 RMSE, 32.43 AP and 70.43 AP50 on test set, which still outperforms both DAVE by 21\%AP, 11\%AP50 and C-DETR by 30\%AP, 28\%AP50. Note that PSECO must be omitted from this comparison since it utilizes SAM for bounding box prediction, thus comparison with GeCo$\_{\overline{\text{Ref}}}$ is neither meaningful nor fair. These results further confirm the effectiveness of GeCo.
>
> >**Include SAM as a baseline counter.**
>
> A simple baseline as the reviewer suggested is not possible, since SAM decoder segments only a single object instance for a given prompt (exemplar bounding box) and is not able to detect multiple objects across the whole image based on the prompt. However, a more advanced baseline that applies SAM with minimal modifications already exists in the literature and is included in Table 1, denoted as SAM-C [1]. This approach is outperformed by all other methods, in particular, GeCo outperforms it by 70\% in MAE and 55\% in AP. This result indicates that the reason for the excellent performance of GeCo is not in the SAM backbone, but rather in the proposed architecture (and the  novel loss function).
>
> [1] Ma, Z., Hong, X., & Shangguan, Q. (2023). Can SAM Count Anything? An Empirical Study on SAM Counting. arXiv [Cs.CV].

---

### Official Review · Reviewer_s2FV · 2024-07-15

**Soundness:** 2
**Presentation:** 3
**Contribution:** 2
**Rating:** 5
**Confidence:** 3

**Summary:**

This paper address the issue for low-shot and zero-shot object counting, with an object detection-based approach. The proposed method heavily uses SAM framework, to provide feature embeddings and refine detection boxes. Attention-based feature aggregation and SAM-HQ are used to get the final features for objectiveness and bounding boxes. Evaluations are carried out on standard datasets with few/one/zero-shot setups, and the proposed GeCo outperforms many state-of-the-art methods.

**Strengths:**

* The proposed method achieves high quality on multiple metrics, including counting accuracy and detection accuracy, and it works well for both low-shot and zero-shot setups.
* The counting loss directly optimizes the detection tasks and greatly improves the model's performance.

**Weaknesses:**

* One concern is that the paper utilizes SAM as its backbone, which is trained on SA-1B dataset. This may leads to some benefits to GeCo by having seeing large amount of diverse data. More discussion is needed to address this.
* Also related to SAM, in table 6, it shows that without detection refinement module, the $GeCo_{\overline{Ref}}$'s performance drops heavily and goes lower than other methods in table 1. I am wondering what if we also feed other detection-based methods' output into SAM decoder. This may be a more fair comparison.

**Questions:**

Does the use of SAM in multiple places bring advantages over other baselines?

**Limitations:**

The limitations are sufficiently discussed in the paper.

---

> ### Author Rebuttal · Authors · 2024-08-06
>
> >**GeCo uses SAM, trained on the SA-1B dataset, as its backbone, potentially benefiting by seeing diverse data.**
>
> Empirical evidence in the Experimental results (Section 4.1) indicates that the key performance gain does not come from the amount of training data in the SAM backbone. For example: the performance of SAM-C, which is a low-shot counting and detection extension of SAM, is significantly lower compared to GeCo (70\%/55\% lower MAE/AP). Another state-of-the-art low-shot counting method, PSECO, uses SAM as well as CLIP backbones (also trained on huge and diverse datasets) but performs significantly worse than GeCo. In particular, GeCo outperforms PSECO by 39\% in MAE on FSCD147. These results strongly support our claim that the excellent performance of GeCo does not come from the SAM backbone, but the proposed architecture and the new training loss. We will emphasize these points in the camera-ready.
>
>
> >**GeCo$\_{\overline{\text{Ref}}}$ performance drops without refinement module. Apply SAM refining also to sota bboxes.**
>
> Note that the refinement module is used to improve **only bounding box accuracy**, not object presence detection, and has in fact a minimal impact on the final count accuracy. This is apparent from that GeCo$\_{\overline{\text{Ref}}}$ achieves a similar counting performance as GeCo (only 7% MAE and 1% RMSE performance drop). A small decrease (2%) in AP50 confirms that GeCo$\_{\overline{\text{Ref}}}$ robustly detects the objects. As the reviewer correctly identified, a larger performance drop (26%) is observed in AP, but this is because the GeCo bounding boxes more accurately fit the detected objects and not because more objects are localized or fewer are missed. The detection performance of GeCo$_{\overline{\text{Ref}}}$ remains superior to the best-performing methods without SAM (e.g., DAVE, C-DETR), where PSECO is omitted as it uses SAM for bounding box prediction.
>
> Additionally, as the reviewer suggested, we feed the most recent state-of-the-art counter DAVE’s predicted bounding boxes as prompts to SAM (DAVE+SAM) and observe improved detection performance by 34\% AP. Note, however, that GeCo still outperforms DAVE+SAM by a large margin (21% in AP and 16% in AP50), which indicates the superiority of the proposed architecture and not the use of SAM. We will add this discussion in the camera-ready.
>
>
> >**Does the use of SAM in multiple places bring advantages over other baselines?**
>
> As demonstrated in our results, GeCo outperforms all state-of-the-art which is also based on SAM (e.g., SAM-C and PSECO), as well as the recent DAVE with SAM refinement step added (DAVE+SAM) – we will add the latter into the camera-ready. The results clearly show that the SAM refinement and the backbone are beneficial, but GeCo still by far surpasses all of the aforementioned sota. We will add this discussion in the camera-ready.

---

> > ### Comment · Reviewer_s2FV · 2024-08-13
> >
> > thanks for the response, which addresses my concerns. I've changed the rating to Borderline accept.

---

### Author Rebuttal · Authors · 2024-08-06

Firstly, we sincerely thank the reviewers for their constructive feedback and hope that responses to your questions clarify the strengths and innovative aspects of GeCo. We appreciate the recognition of novelty and state-of-the-art counting and detection performance. In the following, we summarize our responses, but detailed comments are discussed with each reviewer separately.

Reviewers pointed out that GeCo's strong performance might be due to SAM backbone, trained on the extensive SA-1B dataset. Experimentally, we demonstrated that the strong SAM-based baseline (denoted by SAM-C) achieves significantly lower performance on FSCD147 compared to GeCo, indicating that a powerful backbone (e.g., SAM) is not enough for excellent few-shot counting and detection results. Moreover, GeCo also outperforms a recent PSECO, that uses SAM and CLIP, combined trained on more training data. In the rebuttal, we also clarify that the SAM-based decoder in GeCo is used to refine detections only, i.e., to improve the bounding box accuracy, and does not significantly impact detection robustness and overall count. These results indicate that the key to GeCo’s state-of-the-art performance are the proposed architecture and loss function, rather than the backbone, amount of training data or decoder.

In the rebuttal, we also clarify that the formulation of prototype construction in GeCo is conceptually different from existing low-shot counting and detection methods. In particular, GeCo transforms the few exemplars into a dense set of location-specific prototypes (one prototype per image location). These prototypes are then used to construct dense local object queries, which are finally decoded into object detections. This process avoids over-generalization and false detections and from this perspective differs from the existing methods. Compared to classical DETR-like counters, which can handle only a limited number of queries, the formulation with dense object queries in GeCo enables excellent performance in highly dense and crowded regions. In addition to the architecture formulation, the new loss avoids the issues related to the construction of Gaussian-based ground truth by training the network to predict response maps on which objects are reliably detected by non-maxima suppression.

Performance on objects with high intra-class variability is already demonstrated on the FSCD-LVIS dataset (Section 4.1) — GeCo outperforms state-of-the-art methods by a significant 178% and 73% in AP and AP50. In addition, we include a figure in the rebuttal PDF, to qualitatively demonstrate GeCo’s low-shot counting and detection capabilities on images with high object visual diversity.

In the rebuttal, we also discuss the dependency on the quality of exemplars of low-shot counters. An exhaustive evaluation of this aspect goes beyond the scope of this paper, but an experiment comparing performance on three- and one-shot scenarios (Section 4.1) indicates that GeCo is much more robust to exemplar diversity and selection than other methods. We also show that GeCo significantly outperforms existing methods in dense scenes and small objects, which further highlights its effectiveness in challenging scenarios.

---

### Decision · Program_Chairs · 2024-09-25

**Decision:**

Accept (poster)

**Comment:**

This paper is about "low-shot counting" of objects. After discussion between reviewers and authors, the paper received two ratings of  "borderline accept" and one of "weak accept". The AC recommends that the paper be accepted.